# Feasibility of a Peer-Led Leisure Time Physical Activity Program for Manual Wheelchair Users Delivered Using a Smartphone

Krista L. Best [1,2], Shane N. Sweet [3], Jaimie F. Borisoff [4,5], Kelly P. Arbour-Nicitopoulos [6] and François Routhier [1,2,*]

1   School of Rehabilitation Sciences, Faculty of Medicine, Université Laval, Quebec City, QC G1V 0A6, Canada; krista.best@fmed.ulaval.ca
2   Centre for Interdisciplinary Research in Rehabilitation and Social Integration, Centre Intégré Universitaire de Santé et de Services Sociaux de la Capitale-Nationale, Quebec City, QC G1M 2S8, Canada
3   Department of Kinesiology and Physical Education, McGill University, Montreal, QC H2W1S4, Canada; shane.sweet@mcgill.ca
4   International Collaboration on Repair Discoveries (ICORD), Vancouver, BC V5Z 1M9, Canada; jaimie_borisoff@bcit.ca
5   Rehabilitation Engineering Design Laboratory, British Columbia Institute of Technology, Burnaby, BC V5G 3H2, Canada
6   Faculty of Kinesiology and Physical Education, University of Toronto, Toronto, ON M5S 3J7, Canada; kelly.arbour@utoronto.ca
*   Correspondence: francois.routhier@rea.ulaval.ca

**Abstract:** Active living lifestyles for wheelchair users (ALLWheel) was developed to improve leisure time physical activity (LTPA). The purpose of this study was to assess the feasibility of the ALLWheel program. In a pilot pre-post design, 12 manual wheelchair users in three Canadian cities completed the ALLWheel program (containing 14 sessions over 10 weeks delivered by a peer using a smartphone). Feasibility indicators were collected for process, resources, management, and intervention—before, during, and after ALLWheel. Exploratory outcomes were collected for LTPA (primary outcome), motivation, self-efficacy, and satisfaction with autonomy support and goal attainment—at baseline, immediately following ALLWheel, and three months later. Feasibility was evaluated using a priori criteria for success (yes/no), and within-subjects comparisons were made to explore the change in exploratory outcomes. The participants were 48.9 ± 15.1 years of age and women (66.7%), and had spinal cord injury (41.7%) or multiple sclerosis (16.7%). Feasibility was achieved in 11 of 14 indicators, with suggestions to consider subjective reports of LTPA as the primary outcome in a future randomized controlled trial to overcome limitations with device-based measures and to use strategies to enhance recruitment. Mild-intensity LTPA and satisfaction with goal attainment improved after the completion of ALLWheel. With minor modifications, it is feasible that ALLWheel can be administered to wheelchair users by a peer using a smartphone.

**Keywords:** leisure time physical activity; manual wheelchair; peers; smartphone; feasibility

## 1. Introduction

Physical activity has been considered a behavior that should be prescribed [1,2]. The physical and psychosocial health benefits of leisure time physical activity (LTPA), defined as physical activity an individual engages in during their free time (e.g., walking/wheeling in the park, playing sports, and exercise) [3], are well documented for the general population and are similar for people with disabilities, such as spinal cord injury (SCI). Examples of these benefits include a reduced risk of chronic disease, improved strength and functioning, a reduced risk of depression and isolation, and improved quality of life [4]. Moreover, the benefits of LTPA may be amplified for people with disabilities who use wheelchairs, as prolonged periods of sitting can exacerbate physical health conditions [5]. Many individuals who use wheelchairs are prone to psychosocial sequalae (e.g., depression; isolation) that

may be alleviated through LTPA [6–8]. However, most people with disabilities who use wheelchairs are not active enough to accrue the health benefits [9], with up to 50% of people with SCI not engaging in any LTPA at all [10].

People with disabilities who use manual wheelchairs (MWC) often find it difficult to start and adhere to LTPA due to environmental and psychosocial barriers [11] (e.g., complex health problems, a lack of accessible facilities, transportation challenges, and financial stress) [12–15]. Additionally, fitness and recreational professionals commonly lack knowledge and self-efficacy for adapting LTPA for people with disabilities [16,17]. Several randomized controlled trials (RCT) have addressed some of the environmental barriers to LTPA, effectively using telephone approaches to enhance LTPA among people with disabilities [18,19]. Recently, telehealth has shown promise for improving LTPA for people with SCI, specifically highlighting the potential benefits of mild-intensity LTPA for people who are new to exercise [20]. Peers have also been suggested to have the potential to influence LTPA through motivational interviewing and action planning, but long-term adherence to LTPA remains limited [21]. It is possible that telehealth and peer mentoring may be able to play a role in targeting mild-intensity PA for people with disabilities.

To further address the LTPA needs of people who use wheelchairs, Best et al. developed a peer-led program called active living lifestyles for wheelchairs users (ALLWheel) [22]. The ALLWheel program engages peers as LTPA coaches and uses smartphones to provide LTPA counseling for people with disabilities who use wheelchairs. Self-determination and social cognitive theories that explain LTPA behavior through complex interactions of psychosocial factors (e.g., motivation, self-efficacy, and autonomy support) have been used in populations of people with disabilities [20,23] and provided the theoretical grounding for the ALLWheel program [23]. Using a smartphone may help to overcome some accessibility and transportation barriers associated with LTPA [24]. Moreover, peers can provide a credible source of information to enhance motivation for LTPA through action and coping planning, and goal progression and monitoring in an empathetic way (i.e., through shared life experiences) [22]. The ALLWheel program focusses predominately on mild-intensity LTPA, which can have significant health benefits for people with disabilities and may be less daunting than starting a heavy-intensity exercise regime [20,25]. Further details on the phases of development of the ALLWheel program have been published [22].

The purpose of this study was to evaluate the feasibility of the ALLWheel program according to indicators of process, resources, management, and intervention [26]. An exploratory evaluation of the influence of the ALLWheel program on device-based LTPA (the primary exploratory outcome), self-reported LTPA, motivation, self-efficacy for overcoming barriers to LTPA, satisfaction of psychological needs for LTPA, and satisfaction with LTPA participation was also conducted. The retention of all outcomes was explored three months after the completion of the ALLWheel program.

## 2. Materials and Methods

### 2.1. Design

The exploratory phase of the Medical Research Council Framework [27] was conducted in three Canadian cities (Quebec, Montreal, and Vancouver) using a single-arm pre-post design previously described by Best et al. [22]. The study was approved by local research ethics boards at each site. Informed consent was obtained from all participants.

### 2.2. Participants and Recruitment

A purposive sample of community-dwelling manual wheelchair (MWC) users was recruited through community partners who provide LTPA services for people with disabilities (e.g., Adaptavie, Viomax, and Spinal Cord Injury British Columbia) and clinicians who work in outpatient rehabilitation programs at each site. A minimum of 10 participants were considered adequate for detecting feasibility [28]. To be included, participants had to be between 18 and 65 years of age; live in the community; use a MWC to participate in LTPA; be able to self-propel a MWC for at least 100 m; not meet the LTPA guidelines

of 90 min per week [29]; and be cognitively able to engage in the ALLWheel intervention (Mini-Mental State Exam score ≥ 25 [30]). Individuals were excluded if they had or anticipated a health condition or procedure that could have contraindicated training (e.g., upper extremity injury; surgery), had a degenerative condition that was expected to progress quickly (e.g., amyotrophic lateral sclerosis), or were concurrently or planning to take part in another LTPA program. Participants were screened using the Physical Activity Readiness Questionnaire and e-PARmed-X+ [31].

### 2.3. Intervention

The ALLWheel program consisted of 14 sessions (30 to 60 min) delivered by a peer coach over a 10-week period. As suggested by Arbour-Nicitopoulos et al., an increased number of contacts (twice per week) was targeted during the first three weeks to establish a rapport between the peer coach and the participant [18].

Four peer coaches who were physically active and who had at least 5 years of experience using a MWC were recruited through existing collaborations with study investigators (at least one at each site). Peer coaches received comprehensive training during two 4 h workshops administered by study investigators (K.L.B., S.N.S.). Training for peer-coaches was conducted in person at each site and consisted of a basic overview of LTPA (e.g., choosing activities; choosing the location), rapport building (e.g., getting to know the participant, expressing empathy, and normalizing feelings), goal setting (e.g., how to set SMART goals), and other motivational strategies based on self-determination and self-efficacy theories (e.g., asking questions about LTPA goals to maximize choices, action and coping planning, providing a rationale for suggestions, and discussing how MWCs may be used to accomplish LTPA). A checklist was completed by the peer coaches during or after each ALLWheel sessions (an example is included in the Supplementary Materials).

Upon the completion of all baseline measures (described below), a research coordinator at each site emailed the ALLWheel manual to participants. The manual included an overview of the ALLWheel program and the peer coach approach, the recommended LTPA participation guidelines (i.e., 20 min twice per week of moderate-to-vigorous physical activity, and strength training activities for each major muscle group 2 days per week [29]), goal setting and monitoring worksheets for each session, and action and coping planning worksheets. The research coordinator at each site emailed or called the peer coach to provide the contact details for each participant. The peer coach then contacted the participant by smartphone using a method of their choice (i.e., video call, voice call, or text), with video being the preferred method for its added value of face-to-face communication [32]. All communications took place in the home or community.

As depicted by Best et al., the ALLWheel program integrated constructs from self-determination theory and social–cognitive theory (e.g., competence, autonomy, relatedness, and self-efficacy) [22], and results from a systematic review and focus group [33] (i.e., the pre-clinical phase of the Medical Research Council framework [27]). Guided by the peer coach checklist and the ALLWheel manual, each session was individualized in accordance with the participant-defined LTPA goals. After initial introductions, the peer coach gave a brief overview of the ALLWheel program and the expectations for the study. The peer coach explained the SMART goal framework, then worked with each participant to define SMART LTPA goals. Participants and peer coaches recorded goals in their respective manuals, and goal progression was monitored by the participant. Participants were encouraged to set new goals as their LTPA goal attainment progressed. The peer coach recapped goals at the end of each session, discussed how to overcome potential obstacles, co-created an action and coping plan with the participants, and helped the participant define their desired forms of social support. The peer coach took note of the participants' goals, which formed the initial discussion for the next session.

*2.4. Data Collection*

Data collection was completed by a trained tester before and after the ALLWheel program. Testers received 2–3 h of training by study investigators (K.L.B., S.N.S.). Sociodemographic and personal information (i.e., age, sex, marital status, education, primary diagnosis, length of time using a MWC, level of depression (Hospital Anxiety and Depression Scale (HADS) [34]), and social support (Interpersonal Support Evaluation List (ISEL) [35]) were collected at baseline (T1). Levels of anxiety or depression were interpreted as follows: 0–7 = normal; 8–10 = mild; 11–15 = moderate; 16 or more = severe [36].

Feasibility indicators for process, resources, management, and intervention were measured during study administration and at the end of the study, following the description of Best et al. [22]. Intervention indicators exploring the potential influence of the ALLWheel program on LTPA and proposed theoretical factors (i.e., primary and secondary outcomes) were collected at baseline (T1), immediately following the ALLWheel program (T2), and 3 months later (T3).

Process indicators. The recruitment rate was documented as the number of participants recruited per month. The consent rate was calculated by dividing the number of individuals who met the inclusion criteria by the number of those who consented. The reasons why eligible individuals were not interested in participating in the study were documented. The retention rate was calculated by dividing the number of participants who completed data collection at T2 and T3 by the number of participants who completed data collection at T1. The perceived benefits of the ALLWheel program were evaluated by measuring participant satisfaction with autonomy support provided by the peer coach using the validated Health Care Climate Questionnaire (HCCQ) [37]. The HCCQ comprises six items about perceived autonomy in LTPA (e.g., 'My peer-coach listened to how I would like to do things regarding my LTPA'; I felt my peer-coach provided me with choices and options about LTPA;) on a 7-point scale ranging from 1 (strongly disagree) to 7 (strongly agree). An average total score was calculated. The HCCQ was collected at mid-intervention, immediately after ALLWheel (T2), and three months later (T3).

Resource indicators. Participant adherence was assessed by tracking the total number of ALLWheel sessions attended, recorded by the peer coaches in the peer coach checklist. The ability to recruit and maintain peer coaches at three sites was assessed by tracking the number of ALLWheel sessions completed by the peer coach. Participant and tester burden was measured by the time to administer study outcomes at T1, T2, and T3 as recorded by the tester. The ability for the study investigators to translate all study materials and for the study to be completed in English and French was based on discussions among study investigators and research coordinators during monthly meetings (i.e., yes or no).

Management indicators. The ability to collect actigraphy at three time points was subjectively evaluated (i.e., yes or no) based on the procedures described below. To ensure the ALLWheel program was administered as intended, the peer coach completed a checklist for each session (and each participant) that had three planned check-ins with study investigators after ALLWheel sessions 1, 6 and 11. Fidelity, defined as adherent and competent delivery of the ALLWheel program, was evaluated by study investigators using the completed peer coach checklists that that outlined important details and components of the ALLWheel.

Intervention indicators. Safety of the ALLWheel program was measured by the number of adverse events that occurred during the ALLWheel program as reported by the peer coach.

Exploratory outcomes of the influence of ALLWheel on LTPA and proposed theoretical factors were gathered to aid in the selection of outcomes for a future randomized controlled trial. The primary exploratory outcome was device-based LTPA measured via actigraphy using a small, non-invasive, and lightweight accelerometer (Actigraph GT3X). Participants wore one actigraph on their non-dominant arm between the elbow and shoulder, and one was placed on the rear wheel of the MWC. Actigraph GT3X contains a multidirectional motion-sensitive accelerometer that integrates information about direction and speed to

produce an electrical current with variable magnitudes and durations, and electrical current data are stored as 'activity counts' [38]. The time between sampling units (epochs) was set at 15 s to allow the greatest sensitivity for mild-intensity LTPA [38]. Participants were given the actigraphs at the baseline assessment. One actigraph was placed inside a small custom-made waterproof box that was installed on the rear wheel of the MWC using tie wraps, and participants were asked to wear the other actigraphs at all times during a 7-day period, except during sleep, bathing/showering, or swimming [39]. Participants recorded the time the actigraph was put on and taken off using a log. The tester obtained the actigraphs and logs from the participants at the end of the 7-day period (either in person or via postage-paid envelopes that were provided to the participants). The preliminary validation of actigraphy for evaluating LTPA intensity in wheelchair users has been documented [40].

Secondary outcomes were collected to explore self-reported LTPA and the proposed theoretical influences of the ALLWheel program (i.e., the psychological determinants of behavior change: motivation, self-efficacy for overcoming barriers to participate in LTPA, satisfaction of the psychological need for LTPA, and satisfaction with LTPA participation) to inform the selection of appropriate outcomes in a future randomized controlled trial.

The LTPA Questionnaire for Adults with SCI (LTPAQ-SCI) was used to assess the self-reported frequency (number of bouts per week) and duration (minutes per day) of mild-, moderate-, and heavy-intensity LTPA (for aerobic and strength activities) during the previous 7 days [41]. Weekly total LTPA in minutes was estimated based on the number of bouts per week multiplied by the number of minutes per day for mild-, moderate- and heavy-intensity LTPA (both aerobic and strength activities). The acceptable reliability and construct validity of the LTPAQ have been documented for adults with SCI who use wheelchairs [42].

The Behavioral Regulation in Exercise Questionnaire 2 (BREQ-2) was used to evaluate the motivation to participate in LTPA [43]. The BREQ-2 comprises five subscales that measure varying degrees of LTPA regulation, including external (e.g., 'I take part in LTPA because my family says I should'), introjected (e.g., 'I feel guilty when I do not participate in LTPA'), identified (e.g., 'It's important to me to be physically active'), integrated (e.g., 'I consider exercise consistent with my values') and intrinsic (e.g., 'I take part in LTPA because it is fun') regulations. An additional subscale assessed amotivation (e.g., 'I think LTPA is a waste of time'). Each subscale contains four items except for introjected regulation, which contains three items. Following the statement, 'Why do you take part in LTPA?', participants responded to each item on a 5-point Likert-type scale, ranging from 0 = not at all true for me to 4 = very true for me. The BREQ-2 has been validated in various populations [44,45].

The LTPA Barrier Self-Efficacy Scale (BSE) was used to assess self-efficacy to overcome salient barriers to LTPA participation (e.g., transportation, weather, and pain). The BSE contains 6 items for which participants rated their level of self-efficacy to participate in LTPA in the presence of obstacles on a six-point Likert scale (0 = not confident at all to 6 = totally confident) [46]. The reliability and validity of the LTPA BSE have been documented in spinal cord injury [47].

The validated Psychological Need Satisfaction in Exercise Scale (PNSES) was used to measure satisfaction of psychological needs for LTPA [48]. The level of agreement with 18 items that reflect how participants may feel when participating in LTPA was rated using a 6-point Likert scale from 1 (false) to 6 (true). A mean score was calculated for sub-scales of autonomy (6 items), competence (6 items), and relatedness (6 items).

The Wheelchair Outcome Measure (WhOM), a validated semi-structured interview, was used to develop and evaluate LTPA goals that required the use of a wheelchair [49]. Participants developed a minimum of 1, and a maximum of 10 goals, and then rated the 'importance' of the LTPA goal (on a scale from 0 to 10) and 'satisfaction' with the current performance of this LTPA (scale from 0 to 10) [50]. The goals identified on the WhOM were shared with the peer coach and incorporated into the ALLWheel program. The WhOM was

scored by multiplying 'importance' by 'satisfaction' and taking an average score out of 100 depending on the number of goals set.

### 2.5. Data Analysis

Means and standard deviations (continuous variables), and frequencies and proportions (categorical variables) were calculated. Feasibility outcomes were treated as binary (i.e., yes; no) with success (yes) indicating the protocol was sufficiently robust to move forward with a future randomized controlled trial with little or no adaptation required and non-success (no) indicating the protocol should be revised before proceeding. A priori parameters for success were previously published (see Multimedia file, Best et al. [22]) with modifications summarized in the results.

For the primary exploratory outcome (actigraphy), data were analyzed for 3 days within a 7-day period (2 weekdays and 1 weekend day)m in which both actigraphs were worn for at least 13 h [51]. Using custom algorithms (MATLAB version: 9.13.0 (R2022b), Natick, MA, USA: The MathWorks Inc.; 2022), raw data were converted into mean (SD) number of bouts of mobility (i.e., a volitional transition between activities when using a MWC), duration of bouts (m), average speed (m/s), maximum speed (m/s), and total distance (m). A bout was calculated as any MWC movement that lasted at least 5 s, had a speed greater than or equal to 0.18 m/s (modified from 0.12 m/s from Sonenblum et al. to enhance the quality of data [52]), and ended when less than 0.76 m was wheeled within 15 s [52].

Parametric assumptions for LTPAQ data were assessed. A Wilcoxon signed-rank test was conducted using SPSS version 26 to compare within-subjects changes in total mild-, moderate- and high-intensity LTPA (LTPAQ) between T1 and T2, with the level of statistical significance at $\alpha < 0.05$. A Wilcoxon signed rank test was also conducted to explore within-subjects changes between T1 and T2 for secondary outcomes of motivation (BREQ-2), self-efficacy for overcoming barriers to LTPA (BSE), the satisfaction of psychological needs for LTPA (PNSES), and satisfaction with LTPA participation (WhOM). A Wilcoxon signed rank test was also conducted to explore whether any changes remained at the 3-month follow-up (T3) for all primary and secondary outcomes, with a non-statistically significant difference between T2 and T3 being indicative of maintenance (i.e., $\alpha > 0.05$).

## 3. Results

Twelve participants were enrolled in the study, with one drop-out at T2 (due to loss of interest in the study) and three drop-outs at T3 (due to lack of availability or loss of contact). The mean age of the participants was $48.9 \pm 15.1$ years. Most participants were unmarried (75%) and women (66.7%), with spinal cord injury/disease (41.7%) or multiple sclerosis (16.7%). Participants had experience with an average of 20.0 years of MWC use (ranging from 1.5 to 43.0 years), used their MWC for more than 5 h per day, and propelled their MWC using two hands. The MWC was used at home (100%), at work (33.3%), at school (25%), and in the community (83.3%). Participants reported a high level of social support with a mean (SD) score of 0.7 (0.8) on the ISEL and low levels of anxiety (6.8 (5.8)) and depression (2.9 (3.3)), as assessed using the HADS.

### 3.1. Feasibility Indicators

Definitions of feasibility indicators, a priori parameters for success, and a summary of results are presented in Table 1. Success was achieved in 11 of the 14 feasibility indicators, with minor modifications suggested to proceed with the randomized controlled trial.

**Table 1.** Description of feasibility indicators, parameters for success, results, and suggested modifications for future RCT (randomized controlled trials).

| Feasibility Indicator | Outcome Measure | Parameter for Success | Results | Feasible | Suggested Modification |
|---|---|---|---|---|---|
| **Process** | | | | | |
| Recruitment rate | Number of participants recruited/time | 2 participants/ month/site | 1.5 participants/month | Yes | |
| Consent rate | % participants consented | <10% subject refusal of eligible participants | 19% | No | Relax parameter for success |
| Retention rate | % participants who completed data collection (T2, T3) | Complete T2 & T3 with ≥80% of participants | T2 = 92% T3 = 75% | Yes No | Consider adherence strategies |
| Perceived benefit | Health care climate (mean, SD) | Mean score of >6 | 6.2 (0.5) | Yes | |
| **Resources** | | | | | |
| Participant compliance | Complete 14 ALLWheel sessions | >85% of participants | 92% | Yes | |
| Peer coach adherence | Recruit/retain peer trainers | Facilitate all sessions | 100% | Yes | |
| Data collection burden | | | | | |
| T1 | Time (mean, SD) | >85% of participants complete ≤ 120 min | 103 (41) min | Yes | |
| T2 | Time (mean, SD) | >85% of participants complete ≤ 90 min | 77 (21) min | Yes | |
| T3 | Time (mean, SD) | >85% of participants complete ≤ 90 min | 57 (15) min | Yes | |
| Translations | Translate and administer study materials in English and French | No issues | 0 issues | Yes | |
| **Management** | | | | | |
| Collect actigraphy | Ability to collect actigraph data at 3 time points | ≥80% of actigraph data, 3 time points, 2 actigraphs | 50% of actigraph data, 3 time points, 2 actigraphs | No | Modify 1° outcome for RCT |
| Intervention fidelity | Peer coach checklist | Peer coach administered ALLWheel as intended (>85%) | 90% of peer coach checklists administered as intended | Yes | |
| **Intervention** | | | | | |
| Safety | Number of adverse events | No adverse events | 0 adverse events | Yes | |

Process indicators. Feasibility was demonstrated for three of the five process indicators. Twelve participants were recruited in 18 months (1.5 per month) with a 19% consent rate. Of 123 people contacted us, 61 did not meet the inclusion criteria (they were too active (*n* = 48), had health situation (*n* = 7), or did not use a MWC (*n* = 6)), and 50 were not interested. Of the 12 individuals who completed T1 assessments, 11 completed T2 (92%) and 9 completed T3 (75%). One participant withdrew from the study after two ALLWheel sessions due to health reasons unrelated to the study. The other two participants did not provide a reason for dropping out. All participants who completed the ALLWheel program reported a high level of perceived autonomy support from their peer coach with mean (SD) scores of 6.4 (0.5) at mid-intervention, 6.5 (0.5) immediately after ALLWheel, and 6.5 (0.5) three months after ALLWheel.

Resource indicators. Feasibility was demonstrated in all six resource indicators. In total, 11 out of 12 participants (92%) completed all 14 of the 14 ALLWheel sessions. Briefly, 7 out of 11 participants completed ALLWheel within the intended 10-week period. How-

ever, due to scheduling challenges (e.g., vacations), four participants required additional time (a maximum of 3 weeks) to complete the ALLWheel sessions.

Four peer coaches with spinal cord injury (two males and two females) were recruited through existing collaborations. While an existing skill set was not a prerequisite, the peer coaches had various skills that may have influenced their competency for providing LTPA counseling (e.g., athletic background, motivational speaking, coaching, and previous peer mentorship experience).

The time to complete data collection was a mean (SD) of 103 (41) minutes at T1, 77 (21) minutes at T2, and 57 (15) minutes at T3. There were no reported issues with translating or administering study materials for testing and training in English or French.

Management indicators. Feasibility was demonstrated in one of two management indicators. There were numerous challenges with collecting actigraphs at all time points, suggesting lack of feasibility as a primary outcome. Although the actigraph that was placed on the rear wheelchair of the MWC posed less problems, there were reports of forgetting to wear the actigraph on the arm, not completing the study log, losing the actigraph, or damage to the actigraph. The ALLWheel protocol was successfully administered at all three sites. Minor issues were addressed at monthly team meetings, such as how to proceed if a participant was going on vacation and planned to miss training sessions. The peer coach checklist was completed by the peer coaches at each site and verified by study investigators. One issue that was raised by peers at all sites was that some participants tended to favor voice calls or texts over video calls. The proportion of video versus voice calls was not documented.

Intervention indicators. There were no adverse events during the ALLWheel program.

### 3.2. Device-Based LTPA

Actigraphs placed on the rear wheel of the MWCs and worn on the arm were collected from 6 of the 12 participants. Descriptive statistics (i.e., mean (SD)) for number of bouts, duration of bouts (mins), speed (m/s), maximum speed (m/s) and total distance (m) for both positions of the actigraph are summarized in Table 2.

**Table 2.** Within-subjects summary scores and change in bouts, length of bouts, average speed (m/s), maximum speed (m/s), and total distance (m) immediately after ALLWheel (T1–T2), and 3 months after ALLWheel (T2–T3).

| Actigraphy; Mean (SD) | T1 MWC | T1 Arm | T2 MWC | T2 Arm | T3 MWC | T3 Arm |
|---|---|---|---|---|---|---|
| Bouts | 186.3 (60.3) | 112.8 (56.4) | 183.9 (51.7) | 111.5 (65.1) | 158.2 (48.3) | 65.8 (40.0) |
| Duration bouts (mins) | 6.8 (12.3) | 8.3 (5.0) | 5.2 (0.8) | 5.9 (0.9) | 9.7 (10.8) | 9.3 (12.2) |
| Average speed (m/s) | 0.19 (0.13) | 0.22 (0.09) | 0.17 (0.02) | 0.18 (0.02) | 0.22 (0.17) | 0.2 (0.17) |
| Max speed (m/s) | 1.34 (0.44) | 1.31 (0.36) | 1.28 (0.20) | 1.13 (0.24) | 1.54 (0.52) | 1.11 (0.74) |
| Total distance (m) | 1380.7 (1085.8) | 988.3 (731.1) | 960.9 (359.7) | 614.0 (339.0) | 1702.0 (2148.8) | 800.1 (1136.7) |

MWC: manual wheelchair.

### 3.3. Self-Reported LTPA

There was a statistically significant increase in self-reported weekly mild-intensity aerobic LTPA from 17.9 (36.5) minutes to 73.0 (55.2) minutes between $T_1$ and $T_2$ that did not change significantly at $T_3$. There were no statistically significant differences in moderate- or heavy-intensity aerobic LTPA, or any statistically significant differences between mild-, moderate-, and heavy-intensity strength LTPA. Summary scores for aerobic and strength LTPA are presented in Table 3.

**Table 3.** Within-subjects summary scores and change in participation in mild-, moderate0, and heavy-intensity aerobic and strength LTPA immediately after ALLWheel ($T_1$–$T_2$), and 3 months after ALLWheel ($T_2$–$T_3$), assessed using the LTPAQ-SCI.

| LTPA (Mins/Week) | $T_1$ Mean (SD) | $T_2$ Mean (SD) | $T_1$–$T_2$ *p*-Value | $T_3$ Mean (SD) | $T_2$–$T_3$ *p*-Value |
|---|---|---|---|---|---|
| **Aerobic** | | | | | |
| Mild intensity | 17.9 (36.5) | 73.0 (55.2) | * <0.001 | 61.0 (96.6) | 0.38 |
| Moderate intensity | 12.1 (18.4) | 105.0 (162.9) | 0.06 | 50.5 (149.8) | 0.08 |
| Heavy intensity | 5.0 (17.3) | 14.5 (25.2) | 0.10 | 0.0 (0.0) | 0.28 |
| **Strength** | | | | | |
| Mild intensity | 28.5 (42.4) | 39.5 (66.9) | 0.43 | 64.4 (137.9) | 0.12 |
| Moderate intensity | 11.3 (21.5) | 10.5 (24.5) | 0.48 | 2.2 (6.7) | 0.11 |
| Heavy intensity | 0.0 (0.0) | 3.5 (11.1) | 0.17 | 0.0 (0.0) | 0.17 |

* Statistical significance $\alpha < 0.05$.

Results of the measures of the psychological factors proposed to influence LTPA are presented in Table 4. There were no statistically significant differences between $T_1$ and $T_2$ in terms of motivation, self-efficacy for overcoming barriers, or satisfaction with the attainment of psychological needs. There was a statistically significant increase in satisfaction with LTPA goals after ALLWheel that did not significantly change 3 months later.

**Table 4.** Within-subjects summary scores and change in motivation, self-efficacy for overcoming barriers to LTPA, satisfaction with the attainment of psychological needs, and satisfaction with LTPA participation immediately after ALLWheel ($T_1$–$T_2$), and 3 months after ALLWheel ($T_2$–$T_3$).

| | $T_1$ Mean (SD) | $T_2$ Mean (SD) | $T_1$–$T_2$ *p*-Value | $T_3$ Mean (SD) | $T_2$–$T_3$ *p*-Value |
|---|---|---|---|---|---|
| **BREQ-2, out of 16** | | | | | |
| Amotivation | 2.6 (4.1) | 3.2 (5.0) | 0.28 | 1.2 (2.4) | 0.18 |
| External regulation | 2.6 (4.4) | 2.5 (5.6) | 0.44 | 1.3 (1.7) | 0.22 |
| Introjected regulation | 4.2 (2.9) | 5.4 (4.4) | 0.19 | 2.4 (2.2) | 0.02 |
| Identified regulation | 10.3 (3.4) | 10.1 (3.6) | 0.47 | 10.0 (3.7) | 0.49 |
| Intrinsic regulation | 11.4 (4.1) | 11.5 (4.4) | 0.44 | 11.2 (4.0) | 0.15 |
| **BSE, out of 7** | 4.2 (1.9) | 4.5 (1.8) | 0.13 | 3.5 (1.9) | 0.01 |
| **PNSES, out of 6** | | | | | |
| Competence | 4.4 (0.9) | 4.5 (0.7) | 0.67 | 4.2 (0.8) | 0.35 |
| Autonomy | 4.8 (0.9) | 5.0 (0.6) | 0.48 | 5.3 (0.8) | 0.31 |
| Relatedness | 4.4 (1.4) | 3.9 (1.7) | 0.40 | 4.6 (1.2) | 0.12 |
| **WhOM, out of 10** | | | | | |
| Satisfaction score | 4.9 (3.3) | 6.8 (2.8) | * <0.01 | 4.8 (3.3) | 0.08 |

* Statistical significance $\alpha < 0.05$.

## 4. Discussion

The results of this study confirm that the ALLWheel program was feasible to be administered to MWC users who live in the community. Following minor modifications to address three of fifteen feasibility issues (consent rate, retention at 3-month follow-up, and collecting actigraphs), findings support conducting a randomized controlled trial to evaluate the efficacy of ALLWheel for improving LTPA among WMC users.

### 4.1. Process

Although it was feasible to recruit 12 participants in an 18-month period, it will be important to maintain such efforts to ensure recruitment in a larger randomized controlled trial, as the recruitment of MWC users poses challenges [53]. Some possible strategies include working closely with clinicians and community organizations at each site to establish

links with MWC users as they are discharged from rehabilitation services, and sharing information with special interest groups facilitated knowledge sharing and the uptake of existing LTPA research opportunities. Such communication will be critical to reinforcing a rapport and collaboration with clinicians and MWC users to enhance recruitment in future trials [54,55]. Moreover, recruitment methods included the identification of participants of previous studies (i.e., a motivated sample that may be oversampled in research), working with clinicians through a database of wheelchair users, and sharing posters and presentations to special interest groups in the community and on social media. Word of mouth among the disability community (i.e., snowball sampling) may also be enhanced though the provision of small incentives for referral-based study enrollment (e.g., a gift card for the participants who refer a friend) [53].

Although the consent rate was lower than anticipated (19%), we suggest relaxing the parameter of success for this feasibility indicator. Given that this was a study focused on changing LTPA behavior, which requires a certain level of motivation for initiation, it is possible that many people were amotivated and just not ready to engage in LTPA [56]. More research is needed to determine the best methods for recruiting MWC users and the motivations behind participating in LTPA research. Qualitative interviews with those who accept and those who decline to participate in LTPA research provide a better understanding of underlying motivations and how to better recruit MWC users for community-based LTPA programs.

Enrolled participants demonstrated motivation to engage in the study, and it is promising that retention was high at T2 (92%) and relatively high at T3 (75%). As described by Nary [53], strategies to enhance retention may include ensuring all participants receive the intervention at some point (e.g., offer the intervention to the control group after the study, or waitlist control groups) and to create an appealing study name (e.g., active living lifestyles for wheelchair users, ALLWheel) and a logo. The name ALLWheel was intended to emphasize that everybody can become more active simply by wheeling. Given that ALL-Wheel could be administered completely at a distance, retention may have been enhanced such that participants would not have experienced transportation issues that commonly pose barriers for people with disabilities to participate in research [14].

Finally, a high level of perceived benefit, as evaluated through perceived autonomy support provided by the peer coach, likely influenced retention at T2 and may explain the higher attrition at T3. Given that the peer coach met with each participant 14 times over 10 weeks, it is likely that retention was enhanced at T2. However, between T2 and T3, there was no contact by the peer coach; thus, motivation to complete study outcome may have diminished.

### 4.2. Resources

The ALLWheel protocol was considered feasible for all six resource indicators. Participant compliance was high (92%), suggesting feasibility and acceptability for a peer-led approach to LTPA counseling using a smartphone for people who use MWCs. Similar to the potential impact that perceived autonomy support may have had on retention, cultivating an autonomy-supportive environment likely influenced participant compliance with all but one participant completing all 14 ALLWheel sessions. Offering a flexible schedule that extended beyond the planned 10-week period also likely influenced participants' compliance in completing all sessions. Listening strategies and a non-judgmental approach used by the peer coach, coupled with the prioritization of participants' goals while considering their personal situation and challenges, have been shown to enhance motivation for LTPA [24]. Moreover, peers who use MWCs have been described as being credible, especially for demonstrating techniques when using a MWC [57,58]. In this way, peer coaching may foster a sense of community, which may have been particularly important during the COVID-19 pandemic [59].

The 'peer-effect' may have also been reciprocal, especially during the pandemic (which overlapped with the study duration), such that successful peer trainer retention may be ex-

plained by the benefits of sharing life experiences and perceiving a sense of community [59]. Moreover, peers were included in the development of the ALLWheel program [33], following an integrated knowledge translation approach that has been shown to enhance likelihood for success [60]. In these ways, it was feasible to recruit and maintain peer coaches for this study, and this may contribute to the sustainability of such peer-led digital approaches to LTPA programs in the future.

The total time to collect data was less than anticipated at all three time points, indicating a low burden of data collection and no concerns for the upcoming randomized controlled trial.

### 4.3. Management

While actigraphy represents an approach to collecting device-based data on wheelchair use, the numerous issues reported and missing data indicated that it may not be the best primary measure for evaluating community-based LTPA in a randomized controlled trial. Reports of forgetting to put the actigraph on the arm and forgetting to complete the logbook posed challenges in terms of missing data and not knowing whether movement was being completed by the person independently or if they were being pushed by another person. Although descriptive statistics were calculated, due to the large amount of missing data, there were discrepancies in actigraph outcomes between the arm and the rear wheel. If the participant was not wearing the actigraph on their arm, it is possible they could have been independently pushing their MWC during LTPA and that the information was not recorded. Furthermore, when participants forgot to complete the logbooks, there was no way to obtain information about independent propulsion versus being pushed (one reason why only installing actigraphs on the MWC poses challenges). Finally, the actigraphs were not pre-programmed with start and stop times when given to the participants, which made it difficult to decipher the start and end time of the 7-day period (i.e., sometimes, actigraphs were returned many days after the intended 7-day period due to logistics and scheduling; therefore, the 7-day intended period was used for the analyses). Given that the management of actigraph data was not feasible in this study, we suggest considering an alternative primary outcome for the randomized controlled trial, such as the LTPA questionnaire.

Intervention fidelity is critical to ensuring the transfer of knowledge in the way it is intended. With 8 h of training, three MWC users were able to administer a LTPA counseling program using a smartphone. The checklists demonstrated that peer coaches followed each of the steps of the ALLWheel program; however, some participants preferred communicating with the peer-coach by voice or text message instead of video. Although ALLWheel was originally intended to be delivered using video communication, a recent meta-analysis supports a blended approach as an effective way to increase LTPA [61]. Adjusting the ALLWheel program in accordance with the participants' requests may have also reinforced compliance. Given the need for increased community-based LTPA services and the existing barriers [24], peer-led approaches may represent one approach to enhancing LTPA program offering. However, trade-offs between maintaining the fidelity of evidence-based programs and maximizing suitability in new contexts should be considered in research among individuals with disabilities [62].

### 4.4. Intervention

Evidence supports that peers can develop competencies to promote LTPA with adequate training [21]. Findings from this study confirm that a peer-led approach using a smartphone is safe for providing LTPA counseling in the community.

Exploratory outcomes for LTPA and the psychological factors proposed to influence LTPA were explored to aid in the preparation of a larger-scale randomized controlled trial. There was a statistically significance difference in mild-intensity LTPA between T1 and T2, which is consistent with the goal of the ALLWheel program to get people moving more. This is a promising result, considering that mild-intensity LTPA was associated with a decrease in secondary complications in SCI individuals using MWCs in a previous

study [63]. Peer coaches were not trained to develop or deliver exercise programs. Instead, peers used motivational techniques to develop and monitor the progress of individual goals (i.e., social–cognitive and self-determination approaches) to motivate people to move more during planned LTPA. It is therefore not surprising that there were no statistically significant changes in moderate- or heavy-intensity LTPA. Changes may be indicative of trends toward improvement (as described in Table 3), which may be confirmed in a sufficiently powered RCT.

There were no statistically significant differences observed in motivation, self-efficacy for overcoming barriers to LTPA, or in satisfaction with the attainment of psychological needs between T2 and T3. LTPA comprises complex behavior with multiple underlying factors. Participants were highly motivated when they started the study with mean (SD) scores of 10.3 (3.4) out of 16 for identified motivation and 11.4 (4.1) for intrinsic motivation. Given the study design, all participants experienced interactions with a peer-coach and received resources to overcome barriers and enhance LTPA participation. Similar to Chemtob et al., who reported the maintenance of motivation in a LTPA intervention for people with spinal cord injury [20], it is not surprising that there was no change in these psychological factors in this study. A randomized controlled trial is needed to better understand potential changes in motivation, self-efficacy, and the attainment of psychological needs. Given the practices of goal setting and progression monitoring, and the creation of action and coping plans included in ALLWheel, it is not surprising that there was a statistically significant increase in satisfaction with the attainment of LTPA goals [64,65]. Future peer-led approaches to LTPA should consider the best-approaches for training peer-coaches for motivational interviewing and goal setting techniques.

### 4.5. Limitations

It is likely that participants recruited into this study were already highly motivated at baseline, making it difficult to explore how the ALLWheel intervention may impact elements of intrinsic and extrinsic motivation, and self-efficacy. The effect of social desirability on self-reported LTPA may have also led to an overestimation and an overreporting of LTPA levels to receive social approval [66,67]. The overestimation may have also been higher at T2 given that the participants at T2 may have perceived a strong social bias due to recent interactions with the peers as compared at T3 when three months would have passed since their last interaction.

### 5. Conclusions

The present study supported the feasibility of ALLWheel as a peer-led LTPA counseling program delivered using smartphones for MWC users. With enhanced recruitment strategies and modifications of the primary outcome, this protocol may be used in a larger randomized controlled trial to evaluate the efficacy of the ALLWheel program on LTPA. More research is needed to better understand how psychological factors of motivation, self-efficacy, and perceived autonomy support influence LTPA uptake and maintenance.

**Supplementary Materials:** The following supporting information can be downloaded at: https://www.mdpi.com/article/10.3390/disabilities4010012/s1, Table S1: ALLWheel peer-trainer checklist to ensure program components were followed.

**Author Contributions:** All authors contributed to the conceptualization and development of the ALLWheel program, as well as the study protocol. K.L.B. and S.N.S. trained data collectors and peer-coaches. K.L.B. and F.R. led recruitment and data collection at the Quebec site, K.L.B. and J.F.B. led recruitment and data collection at the Vancouver site, and S.N.S. led recruitment and data collection at the Montreal site. K.L.B. coordinated the team, conducted data analyses, and wrote the first version of the manuscript. All authors have read and agreed to the published version of the manuscript.

**Funding:** This project contributed to the research of the Canadian Disability Participation Project, supported by the Social Sciences and Humanities Research Council of Canada (Grant # 895-2013-1021)

and was funded by the Quebec Health Research Funds; Consortium pour le développement de la recherche en traumatologie/Volet 2 (FRQS: 36548). Salary support was provided to Krista Best by Craig H Neilsen Foundation and the FRQS, to François Routhier by the FRQS, to Shane Sweet by Canadian Research Chair in Participation, Well-Being, and Physical Disability, and to Jaimie Borisoff by the Canadian Research Chair in Rehabilitation Engineering Design at the British Columbia Institute of Technology.

**Institutional Review Board Statement:** The study was conducted according to the guidelines of the Declaration of Helsinki, and was approved by the Ethics Committee of the Centre intégré universitaire de santé et de services sociaux de la Capitale-Nationale (CIUSSS-CN); (2016-541, RIS_2015-467, 17 December 2023) and Research Ethics Boards at the University of British Columbia, Vancouver Costal Health.

**Informed Consent Statement:** Informed consent was obtained from all subjects involved in the study.

**Data Availability Statement:** The data presented in this study are available on request from the corresponding author.

**Acknowledgments:** The authors would like to thank Rainer Molla, Émilie Lacroix, Angie Wong, and Johanne Mattie for their assistance in administering the study protocol and translating documents between French and English. The ALLWheel program and our continuing research have been possible thanks to the contributions and imperative role of the peer coaches in Quebec City, Montreal, and Vancouver (Serge Côté, Marie-Hélène Lapointe-Veilleux, Sherry Craig, and Richard Peter).

**Conflicts of Interest:** The authors declare no conflicts of interest.

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
