# Peer review of "Feasibility of a Peer-Led Leisure Time Physical Activity Program for Manual Wheelchair Users Delivered Using a Smartphone"

_disabilities, doi:10.3390/disabilities4010012_

Round 1
Reviewer 1 Report
Comments and Suggestions for Authors
Congratulations on a careful and through piece of research. See attached for some minor suggestions.

Comments on the Quality of English LanguageSee attached file.
Author Response
Please find our responses to reviewers' comments attached.

Reviewer 2 Report
Comments and Suggestions for Authors
-
Overall, the manuscript is well written and describes a well-structured research protocol with a high translational potential.
-
Some parts of the introduction section could be reduced in length to keep the manuscript concise.
-
Page 2, lines 93 and 94: “using a single arm pre-93 post design as previously described by Best et al.” It may be beneficial for the reader to read a brief description of the design methods and results from the previous research referenced here.
-
It was unclear which primary setting was used by the research participants i.e. residential or institutional while completing activities part of the protocol.
-
Although the participants were recruited using convenience sampling, if available, some of these factors may help readers understand the sample better- the length of their wheelchair usage, time since their injury, their level of physical activity/tendency to exercise regularly at baseline.
-
It was unclear if peer coaches were specifically recruited for this study (and how) or were they already identified expert coaches who had participated in prior research or already coach in some capacity.
Minor comments:
-
Since the study was impacted by COVID-19 pandemic, if the T1 – T2 duration was consistent for all participants.
-
The study involved translating materials between French to English. It was unclear if the researchers had a process in place to ensure that the translations were performed appropriately and consistently by all coaches.
-
Page 12 of 16: Discussion section: Management, Line 512: The word “about” should read “amount”.
Author Response

(The authors gave the same response as above.)
